# 5G NPN Performance Evaluation for I4.0 Environments

**Michail-Alexandros Kourtis** [1,*]**, Andreas Oikonomakis** [1]**, Dimitris Santorinaios** [1]**, Themis Anagnostopoulos** [1]**, Giorgios Xilouris** [1]**, Anastasios Kourtis** [1]**, Ioannis Chochliouros** [2] **and Charilaos Zarakovitis** [1]

1 National Center of Scientific Research "Demokritos" 1, 15341 Agia Paraskevi, Greece
2 Hellenic Telecommunications Organization (OTE) S.A, 15124 Marousi, Greece
* Correspondence: akis.kourtis@iit.demokritos.gr; Tel.: +30-21-06503109

**Abstract:** This paper aims to develop an open Asset Administration Shell (AAS) solution for 5G Non-Public Network (NPN) management, focusing on manufacturing digitization and complete Information and Operational Technology (IT/OT) convergence. The proposed 5G NPN framework is evaluated in a factory-like simulation scenario considering network slicing for I4.0, and demonstrates the outlook of 5G communication in the industrial domain, achieving an upload data rate of up to 86 Mbps, and a Round-Trip Time (RTT) for end-to-end communication as low as 11 ms. The proposed framework integrates OPC UA as an enabler and middleware across different protocols, equipment, and the manufacturing shop floor, with the target of aggregating different industrial data and creating insights on production optimization in a unified manner. The framework combines 5G NPNs with I4.0 environments, in the form of a complete FNMS and its corresponding AAS. In parallel, a set of I4.0 enablers are investigated within the framework of the project, covering a Time-Sensitive Network (TSN) on the shop floor. The main objective of this paper is to propose a method for the unified integration of various enablers in the I4.0 domain and their combination with 5G technology, and to evaluate the feasibility of hosting industrial applications and services over 5G channels through the implementation of different slicing schemas. The paper presents detailed experimental data regarding 5G downlink/uplink data rates and RTT delays.

**Keywords:** I4.0; 5G; NPN; OPC UA; network slicing





## 1. Introduction

The emergence of twin transition paradigms in multiple technological fields with the introduction of 5G telecommunication standards in recent years [1] has led to a rapid convergence between the telco and industrial domains [2]. Industrial domains and factory spaces include a large variety of heterogeneous devices with stringent communication requirements and complex topologies involving different processes. This renders the use of public telco networks (i.e., public 5G infrastructure) challenging, as factory floor operators usually require full control of the underlying network and total provision of the corresponding resources [3]. This has led to the introduction of 5G Non-Public Networks (NPNs) for industry protocols and operations; however, the evolutionary shape of 5G networks and their modular architecture requires an equal plug-and-play functionality in each environment for proper adaptation. The Reference Architectural Model Industry 4.0 (RAMI 4.0) [4] paradigm defines such a modular factory environment, and sets a solid groundwork for the further adaptation of 5G networks to the industry floor. I4.0 has opened up various fields and technological candidates to drive innovation and progress [5,6]; in terms of communication, 5G meets the necessary requirements to support such a demanding environment.

From the first Industrial Revolution (I1.0), where the introduction of water- and steam-powered machines helped workers to carry out their tasks, the increased efficiency and capability necessitated growth in other areas—businesses grew from individual interests to organizations with owners, managers, and employees serving customers. In I2.0, electricity

replaced water and steam as the primary power source, and enabled the concentration of power sources to fewer machines and, eventually, machines having their own power sources. Paralleling mechanical innovation was the innovation in workforce organization when I3.0 occurred, in the last few decades of the 20th century, known as the first computer era. The invention of devices such as the transistor and, later, the integrated circuit chip made it possible to fully automate machines to further aid or entirely replace human operators. Software systems were developed to make complete use of the new possibilities created by electronic hardware. Integrated manufacturing systems were replaced by enterprise-scale planning tools, which allowed users to plan, schedule, and track product flows.

This convergence into I4.0 must also address a wider set of topics related to real-time response problems, energy efficiency, performance, privacy, and security—topics covered in detail previously [7]. Within the scope of this paper, 5G NPN deployments are used to evaluate the real-time response criticality of I4.0, 5G, and performance. Nonetheless, security and privacy are vital components of every 5G NPN setup, and the proposed integration with IDS primitives addresses this aspect partially in the proposed work.

5G has already been applied in industry-related scenarios and setups [6–12], mainly focusing on wireless communication acceleration and multipoint connectivity—features that can essentially boost factory floor operations in a seamless manner. Apart from the use of 5G fast-path communications to replace wiring [10,11], the I4.0 vision outlines how unified automation and holistic provision can also push forward innovation in industrial fields [12–14]. The key enabler in this unified namespace evolution of I4.0 is OPC UA [14]. OPC UA is a unified industrial component dictionary and communication specification that aims to combine different vendor protocols and prototypes in a homogeneous manner for industrial Asset Administration Shell (AAS) systems. This manuscript proposes a hybrid OPC UA and 5G convergence to a prototype AAS for I4.0 as a stepping stone for 5G communications on the plant floor. This proposal forms a prototype architecture combining different technologies and access technologies that can benefit I4.0, with 5G as the key enabler.

Another key aspect of this work is the integration of a TSN [15–17] as a functional component in the I4.0 environment, through the 5G network slicing paradigm. TSNs, as a specification, define strict time and priority requirements, conforming to the demands of the factory floor. 5G slicing, on the other hand, offers a multipoint, fast-path communication channel and, using the appropriate configuration on Radio Access Network (RAN) provision, the transmission of TSN-aware communication. Within the framework of this paper, a set of experimental results investigating the 5G network slicing capacity for bandwidth (downlink, uplink) and latency are presented, in order to ensure current system standards in terms of TSN facilitation. The experiments were performed in a real-world 5G NPN factory simulation network.

Finally, AI can play a pivotal role in the seamless integration of these heterogeneous fields, being able to improve and optimize specific vertical domains in the manufacturing industry [18]. In the framework of Zero-Defect Manufacturing (ZDM), AI can be used to minimize error defects and detect challenging production anomalies, in order to optimize high-cost part manufacturing [19]. In the field of energy, production material efficiency and reducing $CO_2$ emissions are crucial factors. The use of AI can help to drastically reduce a production line's $CO_2$ emissions, as well as rendering it as environmentally circular as possible. Finally, for cybersecurity, it is crucial to assure protection of every industrial asset [20]. The use of data processing acceleration techniques (e.g., in-network computing and programmable ASICs) and novel communication paradigms (e.g., OPC UA) allows us to address cybersecurity at different levels, providing the basis for a secure I4.0 [21].

The remainder of this paper is organized as follows: Section 2 presents various domains in which I4.0 and 5G can be combined, and describes how this can be mutually beneficial. Section 3 presents an I4.0 architectural proposal based on the combination of different technologies, and details how the different aspects of each are leveraged. Section 4

describes a set of preliminary experimental tests considering 5G network slicing on a prototype 5G NPN factory floor, and outlines the advantages of 5G. Finally, Section 5 concludes the manuscript, and provides future directions for related research.

## 2. I4.0 and 5G Convergence Background

This section covers the different enabling technologies of I4.0 and 5G convergence in discrete, thematic sections, namely, 5G in the factory space, cloud-to-edge continuum, data sharing, and governance and plant floor openness. An overall schema of this process is depicted in Figure 1.

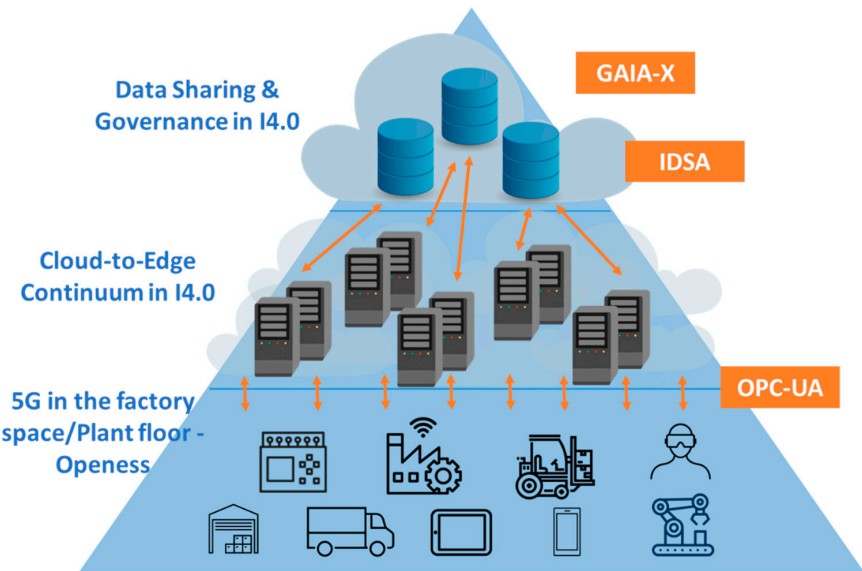

**Figure 1.** I4.0 and 5G convergence background.

5G in the factory space: Ongoing digital transformation is essential for the manufacturing industry to transition towards the I4.0 vision [22]. Such transformation is being implemented across all industrial domains, including communications, which remain as the backbone for seamless operations and data exchange across devices and services. The coexistence of existing TSNs and new 5G URLLC capabilities can be seen as a key enabler for future smart industrial processes, where meeting real-time requirements is crucial [23]. However, various technological gaps in the full integration of 5G and TSNs have been identified [19], with the most significant being that of deterministic data transmission and QoS support, which remains an open challenge [24]. The proposed framework includes a novel mapping from the TSN mechanism to the 5G domain by leveraging slicing mechanisms from the NPN perspective. In the I4.0 ecosystem, the use of an NPN allows a vertical to have an end-to-end in-premise 5G network, such that the private traffic can be confined within the boundaries of the defined premises, without the need to reach the public domain.

Cloud-to-edge continuum in Industry 4.0: Modern manufacturing industry practices have invested heavily into data transport from the factory floor to the cloud, where bulk processes and monitoring data are analyzed in depth, leveraging the computing superiority and resource flexibility of a remote cloud infrastructure [25]. However, cloud processing renders the trustworthy exchange of information between different industrial data providers in a certified manner difficult. Moreover, considering the heterogeneity of data sources and the requirements of offered services found in Industry 4.0 scenarios, different processing techniques must be considered. In this context, the use of different processing paradigms, such as edge computing, stands as a possible solution for those industrial processes with real-time requirements. 5G integration with an I4.0 edge can offer turnkey functionalities, dynamic asset management, and the necessary interfaces to onboard, test, and validate various innovative apps, while ensuring uninterrupted system operations

and availability [26]. This can be achieved in two stages: (i) the digital twinning of the operational protocols present on the factory floor (i.e., PLC, ERP, MES, and SCADA) by following an OT-as-a-Service (OTaaS) approach based on soft sensors, and creating a validation framework for the various new services to be audited; and then (ii) the secure deployment of the developed services on the factory floor, making use of different processing and storage techniques (depending on the needs), and initiating continuous monitoring of the OT process chain.

Data sharing and governance in I4.0: Data sharing already plays a crucial role in the manufacturing industry; either through data sharing for defect detection, or through entire production chain provision and analysis, a significant amount of data is generated each second on the factory floor [27]. Existing horizontal approaches cannot encapsulate the relationships between the different domain processes, actors, and equipment in the OT environment, such that each domain analysis remains isolated and monolithic. Additionally, the rapidly developing EU manufacturing industry horizon ratifies the collaboration and data exchange between different industrial enterprises for mutual benefit and growth. However, the sensitive nature of industrial data necessitates the formation of a trustworthy and sovereign data-exchange mechanism. In this paper, we propose a holistic solution for data management in the manufacturing context, which covers the different layers of the value production chain, following the RAMI 4.0 architecture [6]. This creates a unified approach for data processing across all domains, where each layer can provide feedback and interact with its counterparts, ensuring aggregated provision of the manufacturing process. The proposed system is based on the International Data Spaces (IDS) [28,29] primitives for identity providers and connectors, which are used to create trust chains that support creation, evaluation, and acceptance of software artefacts by creating individual signatures after each step (i.e., publication and evaluation stages). This ensures that the data that have been agreed on to be used will be deleted immediately after they have been used (Clearing House), thus ensuring trust among parties.

Plant floor openness: The drive to create a common foundational technology for automation in the I4.0 manufacturing space has created the core challenge of a vendor-independent open-interface platform for production enterprises. While different standards have been established at the IT and OT levels, they continue to operate in an isolated or partially integrated manner. Open communication, open interfacing, and open specifications are integral parts in the pathway towards an I4.0 FoF. The proposed architecture leverages the Process Automation Device Information Model (PA-DIM) of OPC UA—currently used to implement the Asset Administration Shell (AAS) for I4.0—and extends it for the provision of the 5G NPN, edge, and TSN-aware infrastructure and equipment [14]. The designed factory floor integrates 5G NPN and TSN primitives directly in the OPC UA ecosystem, and provides the necessary interfaces for control, data collection, and management. At present, OPC UA and 5G are separated or partially integrated in the manufacturing ecosystem. They use different management interfaces for their control and management; thus, different operator training is required. Our ultimate aim is to design a unified interface, through the use of a federation layer, in order to provision and manage both entities seamlessly. This can be achieved through exposure of the necessary 5G NEF functions related to I4.0—defined by 3GPP—to a PA-DIM entity.

## 3. Hybrid 5G and I4.0 Concept Architecture

The centerpiece of the proposed framework is the reference architecture depicted in Figure 1, exploring a wide set of solutions and technologies for data-driven distributed manufacturing spaces. The architecture is split among different layers covering all aspects of data management and processing in the industrial ecosystem, namely, (i) the federation of different data spaces; (ii) a unified AAS for I4.0 and 5G components; (iii) an FNMS provisioning the lifecycle, monitoring, and security of factory networking; (iv) AI enablers for I4.0 components covering both OT and IT infrastructure and data; (v) provision of Field-Level Communications (FLC) at the factory floor; and (vi) the development of data-centric

manufacturing verticals. The proposed system is based on the integration of a 5G NPN with I4.0, in the form of a complete FNMS and its corresponding AAS. In parallel, it focuses on implementing a cohesive and trustworthy data management and federation framework for efficient industrial data exchange.

### 3.1. Data Space Federation

The federation of data spaces focuses on the end-to-end management, handling, and governance of the data generated across different industrial spaces and exchanged between different providers. This architectural layer combines the GAIA-X and IDSA ecosystems, in order to provide a holistic approach to data processing and management in a secure and trusted manner, covering both the infrastructure side (GAIA-X) and the core data side (IDSA). Furthermore, the overall industrial data management for resource identification can be aligned with the IDS-RAM, based on the ISO/IEC 2382:201513 (Information technology—Vocabulary) standard. This can establish 5G convergence to I4.0 standards as a fully compliant solution with the general I4.0 directives, while enabling seamless integration with IDS specifications for both the OT and IT components of the framework. Furthermore, adoption of this specification enables the extension of the proposed AAS to 5G infrastructure, and facilitates the full automation of NPNs in the manufacturing space.

### 3.2. 5G-Enabled AAS

AASs already play a focal role in the I4.0 ecosystem, and serve as the catalyst for full adoption of the proposed enhanced technologies in the manufacturing field, along with interoperability across value chains. The proposed AAS solution can provide control and provision functions for both OT and IT equipment and modules. Furthermore, an important aspect in the implementation is the unified management of the lifecycle of both domains in a seamless manner. The architecture takes into account the asset lifecycle stage, according to IEC 62890, as considered in RAMI 4.0. Such lifecycle orchestration considers the heterogeneity of various physical and virtualized devices and services, and focuses on building the tools required for seamless commission, re-configuration, and decommission. Due to the variety of underlying technologies, a multifaceted AAS framework is implemented, which incorporates separate AASs in a many-to-one relationship. The distinct sub-AASs are the industrial AAS (for various equipment and services), a 5G UE AAS (including the interfaces for 5G-enabled equipment), and the 5G network AAS, supporting the core, RAN, and TSN components. Maintenance and management functions for the 5G-centric operations are performed by the 5G-OAM subsystem, also called the 5G operation support system (5G-OSS).

The main points of innovation related to the 5G Network AAS, as depicted in Figure 2, are (i) upgrading to the latest 3GPP release, which adds new features to the UPF functions and, to some extent, redefines the QoS; and (ii) adding new features (e.g., localization services such as those being studied in 3GPP SA2 for Release 17, "Enhancement to the 5GC location services phase 2 (5G_eLCS_ph2)"). Additional changes that the framework can support during its operation include (i) reconfiguration of communication links, such as QoS or addition of new 5G connections, while updating the 5G network AAS and related entries in the 5G UE AAS; (ii) deployment of different UPF functions on different host computers to boost capacity and redundancy; and (iii) flexible communication across the hierarchies of a factory using 5G network slices. This means that AAS information can be updated throughout the factory's lifecycle (i.e., across development, engineering, operations, and maintenance, to final scrapping and recycling).

The crucial aspect in this core objective that the proposed prototype addresses is linking the different OT and IT entities, in order to operate them in a universal and technology-agnostic manner. To achieve this, the OPC 30081: OPC UA for Process Automation Devices (PA-DIM) model can be utilized, in order to create a ubiquitous asset reference model for OPC UA compatibility. PA-DIM can be extended appropriately to support 5G ontologies and create a holistic I4.0 information model. The industrial AAS and the 5G AASs are able

to exchange information through the APIs that utilize this meta-model. The main objective of the proposed AAS solution is to provide a universal factory network management system (FNMS) based on I4AAS—the OPC 30270: UA for Industry 4.0 Asset Administration Shell specifications and the industry standard IEC 62443.

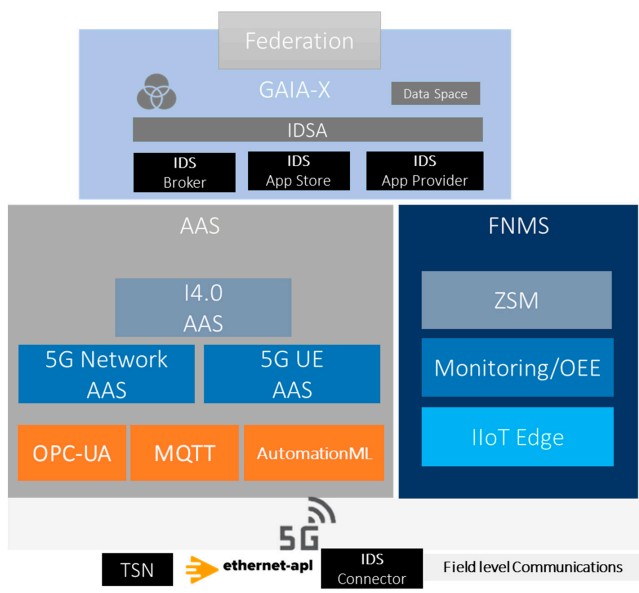

**Figure 2.** Proposed architecture for I4.0 and 5G convergence.

Figure 3 provides a brief overview of the possible interfaces between the 5G AAS and the 5G system. The 5G system consists of various physical and logical components; the 5G core network controls the data flows and connections between the 5G device endpoint (the UE) and the Data Network (DN) endpoint (the user plane functions or UPFs), and the 5G Network Management System (NMS)—which has a service-based architecture—is responsible for configuring the physical and logical nodes of the 5G core network domain.

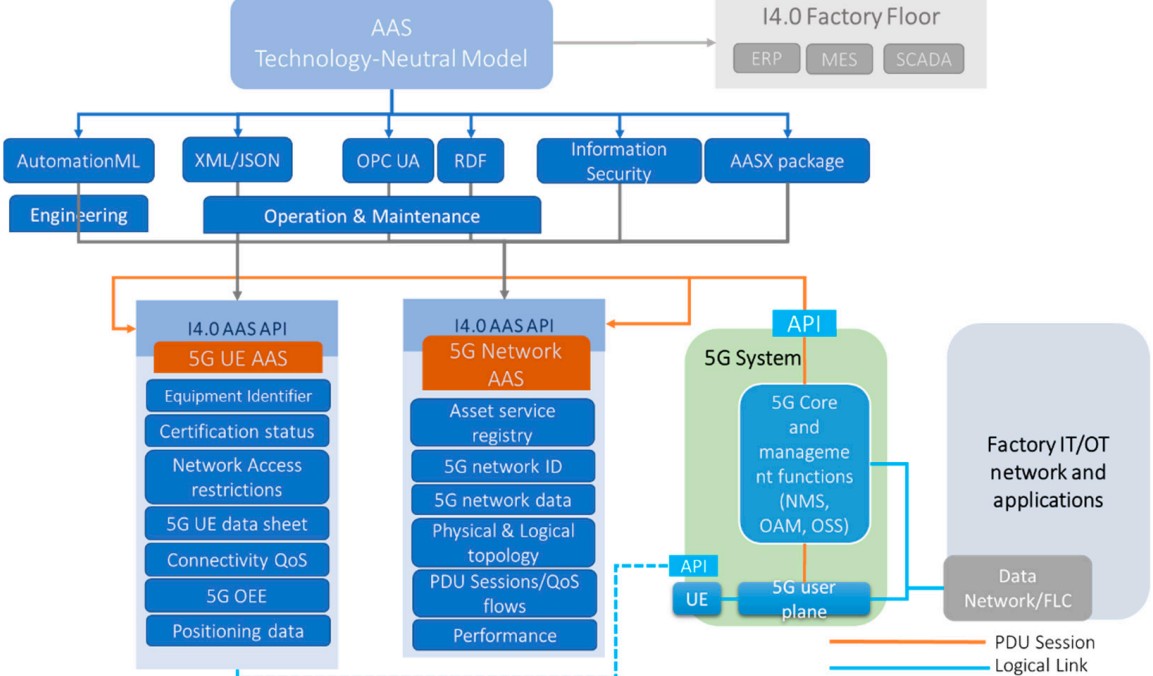

**Figure 3.** AAS support for 5G networks.

The 5G system is able to communicate the network status and events, as well as obtaining external configurations through a standardized Application Protocol Interface (API). It enables connections via a northbound interface to an external system, and through a southbound interface to the various 5G network functionalities. The 5G system and I4.0 AAS are able to exchange information by means of APIs. A specific API must be added to manage the information in the 5G network AAS and the 5G UE AAS. Both of these use standardized data models and enable uniform access to all relevant data across all lifecycle phases, independent of the specific representation models used (e.g., AutomationML [30] for engineering, or OPC UA [5] for operations and maintenance).

In future versions of the 5G network AAS, the 5G system can be configured by AAS logic, enabled by an "active AAS" with integrated smart functions. To support interactions between the AASs of different entities, the 5G network AAS and 5G UE AAS must be able to talk in an I4.0-specific "language". The AAS began as a basic information pool, consisting of simple files that stored relevant information in a highly reliable manner in an agreed data structure—for example, using JSON description models.

A virtualized representation of all components, together with an AAS description of all relevant radio network components, provides an elegant way to encapsulate the complexity of a 5G system. Two special cases require further analysis: when a 5G UE is first integrated into a manufactured product, such as an engine, all of its properties must be described in its AAS. This also includes radio parameters if the UPF functionality can be implemented, either in a standalone device or in combination with an industrial gateway that includes a firewall.

### 3.3. 5G and I4.0 FNMS

The management of 5G on the factory floor requires resource isolation and flexible turnkey physical/virtualized entity management. Therefore, a Non-Public Network (NPN) is the preferred solution, which enables the deployment of a 5G system for private use, simplifying on-the-fly configuration and ensuring resource allocation exclusivity. In the framework of the proposed FNMS, the NPN can be deployed as a Standalone Non-Public Network (SNPN), which is operated by an NPN operator and does not rely on network functions provided by a PLMN. An SNPN is identified by a combination of PLMN ID and NID (Network Identifier); however, 5G NPNs are not limited to the core and RAN parts—they also need to address the routing, onboarding, and commissioning of various industrial devices that are not fully integrated with the 5G specifications. A 5G FNMS can address this open challenge by integrating a 5G LAN-type service, which provides services with similar functionalities to Local Area Networks (LANs) and VPNs, but with improved 5G capabilities (e.g., high-performance and long-distance access, mobility, and security). The 5G LAN-type service enables management of 5G Virtual Network (VN) group identification, membership, and group data. The Network Exposure Function (NEF) also exposes services to dynamically manage 5G VN group data. Furthermore, 5G supports optimized routing by enabling support for local switching at the UPF, without having to traverse the data network for UE–UE communication when two UEs are served by the same User Plane Function. In the framework of the proposed architecture, the OPC UA specification for interfaces and add-ons (OPC 10001-7) can be used as a basis to interlink the 3GPP NEFs. This can achieve full I4.0 integration of 5G exposure primitives, and automated management and configuration through an OPC UA-compliant interface.

### 3.4. Monitoring

The monitoring component refers to the supervision and observability of the entire data space infrastructure, covering both OT and IT resources. Manufacturing domain monitoring focuses on two aspects of the infrastructure, and aims to improve the data processing of raw machining data. The first aspect addresses the manufacturing process optimization, targeting defect minimization and significant energy reduction, while the second aspect concentrates on predictive maintenance of the machining tools, which

can also greatly contribute to OPEX reduction and equipment effectiveness. As optimal production is pivotal in the age of I4.0, Overall Equipment Effectiveness (OEE) has emerged as a Key Performance Indicator (KPI) in the manufacturing industry. OEE benchmarks the percentage of manufacturing time that is truly productive. An OEE score of 100% indicates that one is manufacturing only good parts, as fast as possible, and with no downtime. The OEE KPI formula incorporates three elements: availability (in terms of downtime), performance (the speed of production), and quality (the percentage of products up to standard). Regarding the industry domain, convergence of 5G and I4.0 focuses on extension of the OPC 40501-1 specification targeted for machine monitoring, integrating it across the process chain, where suitable translators can be implemented per demand. Subsequently, for the 5G NPN domain, an I4.0 framework can follow and integrate the directives of the Network Data Analytics Function (NWDAF) 3GPP specification (33.521). A 5G monitoring solution can be a purpose-built standalone NWDAF, in order to provide a fully standards-based 5G analytics capability. Not merely an API connection to a central analytics platform, but direct integration to an AI-enabled edge platform for intelligent data processing, can provide real-time feedback on the 5G IT and OT states, thus offering practical insights on the factory floor.

### 3.5. IIoT Edge

Information systems that support production processes (e.g., ERP, MES, SCADA) rely on data collected from the shop floor. In many cases, these data are still stored locally and are integrated over the infrastructure manually; however, automated solutions that allow for data integration are tending to become a common solution in practice. Until recently, only non-standard data collection systems designed for specific problems were available on the market. In case of larger production compositions, where various different machines and controllers were used, custom system integration solutions had to be developed in order to integrate all data. With the emergence of the I4.0 initiative, significant progress has been achieved in converging IT and OT environments. Data collection software now uses web protocols and protocols for Industrial Ethernet (e.g., Profinet, OPC UA) in order to display and manipulate data and manage IoT devices, significantly easing data collection efforts while providing rapid adaptability and scalability. All of this enables factory automation solutions that are more intelligent, distributed, collaborative, and easier to integrate. The I4.0 edge can offer turnkey functionalities, dynamic asset management, and the necessary interfaces to onboard, test, and validate various innovative apps, while ensuring uninterrupted system operations and availability. This holistic system and its extensions can achieve (i) digital twinning of the operational protocols present on the factory floor (i.e., PLC, ERP, MES, SCADA) following an OT-as-a-Service (OTaaS) approach, and creating a validation framework for the various new services to be audited; and (ii) secure deployment of the developed services on the factory floor, initiating continuous monitoring of the OT process chain.

### 3.6. Field-Level Communications

In this framework, Field-Level Communications (FLCs) are regarded as a pivotal factor in intra- and inter-factory connectivity, as well as a vital aspect in the integration of 5G NPNs in the manufacturing ecosystem. The presented approach to FLCs aims to address the following open challenges: (i) OPC UA [31] application convergence, where multiple OPC UA automation devices from multiple vendors must share one network; (ii) OT convergence [32], where multiple systems and devices from multiple vendors using different OT protocols must share one network; (iii) IT/OT convergence, where multiple controllers, devices, applications, and systems from different vendors using a combination of IT and OT protocols must share one network; (iv) IT/OT organizational convergence, where the boundary between organizations is blurred, and management of IT and OT groups operates under common strategies and processes; and (v) the evolution to wireless communications, for which the most recent advances in 5G technologies (specifically

those related to URLLC and mMTC) are crucial. The implementation is to be based on two key technologies: TSNs, and Ethernet APL. TSNs, according to IEEE 802.1, support communications with bounded latency and jitter—a critical feature in manufacturing operations. Ethernet APL, on the other hand, delivers seamless Ethernet connectivity to sensors and actuators in process automation, including hazardous areas. This technology convergence focuses on delivering a solution for controller-to-device communications, according to the IEC 61784 profiles for interoperability in factory manufacturing and process control. Another important aspect in the industrial environment is safety; FLC capabilities can leverage an OPIC-10000-15-compliant framework, including the necessary processes and mechanisms for functional safety, as defined in the IEC 61508 and IEC 61784-3 series of standards. This includes the implementation of an application-independent component, where standard factory communications and additional safety functional entities can be integrated.

### 3.7. TSN

5G, the next generation of 3GPP technologies, offers capabilities specifically designed to meet certain industrial needs. These include URLLC in 5G-NR (New Radio), support for TSNs, and network deployment scenarios for Non-Public Network (NPN) operations, ranging from standalone NPNs to public-network-integrated NPNs. In Release 16, 3GPP adopted 5G-TSN [33] integration for time-sensitive communication. This framework focuses on 3GPP standard adoption from Release 17 and onwards, with respect to TSNs, and fosters the integration of relevant industrial automation profiles for TSN activities, as defined in IEC/IEEE 60802. The manufacturing industries involved have requirements associated with guaranteed high availability, high throughput, real-time transmission, low latency, and low jitter, and currently utilize vendor-locked protocols and solutions, such as Profibus, Profinet, and EtherCAT. To overcome this open challenge, we provide a vendor-independent TSN solution over 5G with OPC UA Pub/Sub compatibility, embracing end-to-end interoperability for every industrial automation application, as detailed in Figure 4. More specifically, this solution can address the following industrial automation traffic types through its proposed 5G-enabled TSN: (i) time synchronization IEEE 802.1AS; (ii) scheduled traffic IEEE 802.1Qbv; (iii) IEEE 802.1Qbu recommended frame pre-emption; (iv) IEEE 802.1Qbu—optional strict priority IEEE 802.1Q; (v) redundancy IEEE 802.1CB; and (vi) TSN configuration IEEE 802.1Qcc.

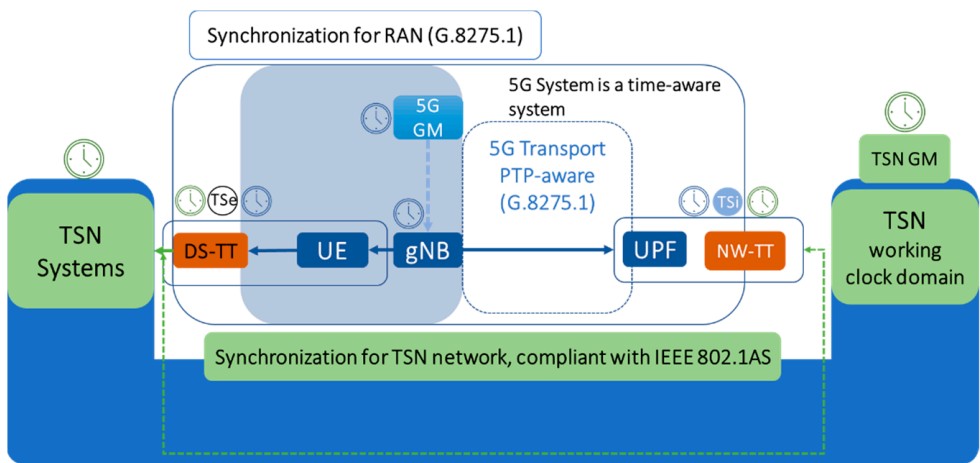

**Figure 4.** TSN support for 5G networks in I4.0.

### 3.8. IDS Connector with OPC UA Integration

IDS connectors are intended to be used for secure data transfer between two or more data endpoints. The connector architecture uses application container management technology to ensure an isolated and secure environment for individual data services. Data

preprocessing is often performed by data apps, which are data services encapsulating data processing and data transformation functionalities. Data apps are bundled as container images for simple installation through application container management.

In many cases, there is a need to integrate IDS connectors with existing back-end systems. To enable this, the connectors must implement specific APIs, in order to either receive or transmit data from or to other systems. On the one hand, OPC UA has been widely used across diverse industrial fields for integrating hardware devices and interconnecting systems, based on a client–server architecture. On the other hand, IDS connectors provide customized interfaces to enable OPC UA servers to notify one another about data node updates through an IDS-based data pipeline. Figure 5 shows an example scenario of OPC UA and IDS connector integration, where two factory production machines are connected to dedicated OPC UA server instances. In addition, there exist specific IDS connectors for OPC UA servers, which implement the actual data exchange between the servers. The software modules that manage all data related to Machine 1 (i.e., OPC UA Server 1 and IDS Connector 1) are deployed in the same security domain. Similarly, data components connected to Machine 2 (OPC UA Server 2 and IDS Connector 2) should be hosted on their own cloud environment or Docker network, for example. Communication between the two separate security domains is implemented using the secure IDS Communication Protocol (IDSCP). The aforementioned architecture specifies a one-way communication scenario between the two OPC UA server installations through IDS connectors. This requires extending IDS Connector 1 with a similar custom container configuration as deployed in IDS Connector 2. Similarly, IDS Connector 2 should be extended with a similar data routing configuration as implemented in IDS Connector 1.

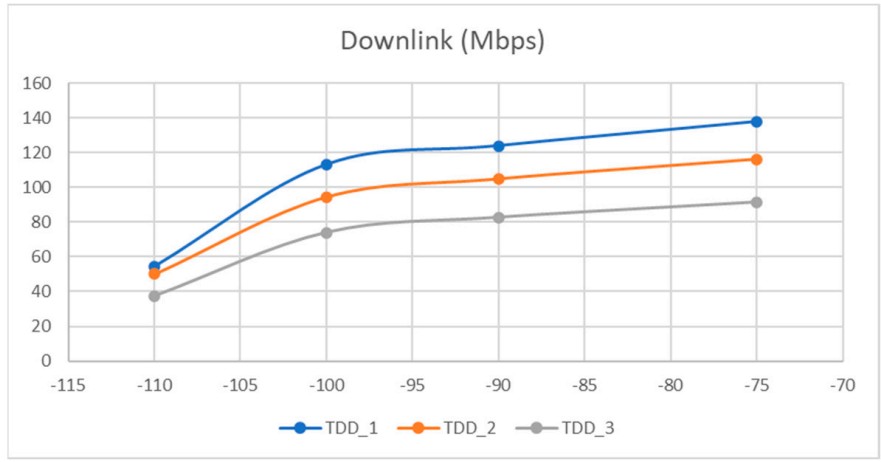

**Figure 5.** Downlink rates for different TDD values.

### 4. I4.0 and 5G Framework Evaluation Measurements

This section presents the 5G NPN performance tests, deployed for a simulated I4.0 scenario. The purpose of these experimental sets was to replicate an industrial communication use-case considering multiple devices and different requirements. The deployed 5G NPN testbed was configured to operate under different configurations with different strict network requirements for gateways and devices. This set of measurements refers to the measurements of uplink, downlink, and RTT of a 5G NPN operating in an I4.0 environment. It is based on an SA 5G network implemented through an AMARISOFT callbox, according to 3GPP Release 16. The AMARISOFT callbox provides a stable 5G mobile network system for testing and experimentation. Both the 5G core and 5G RAN (gNodeB) functions were software-defined and were run on an x86 Linux-based host. The 5G RAN operated PCIe SDR boards as an RF front-end. The system used for the measurements supports TDD transmission mode, with the ability to allocate different slots for the uplink and downlink channels. The bandwidth was 40 MHz, and the supported modulation schemes ranged

up to 256 QAM for the downlink transmission channel and 64 QAM for uplink. The 5G terminals were commercial 5G mobile phones.

The flexibility in the Time-Division Duplex (TDD) mode of operation makes 5G ideal for communications on the factory floor, as the allocation of different time slots in the uplink and downlink may fulfill the requirements for the various equipment present. TDD in 5G is commonly used in scenarios where the uplink and downlink data rates are asymmetrical, and its most common deployment type is in dense topologies with low-power nodes [34]. This network deployment better supports applications that include positioning, mobility, and multicast communications. Regarding different slicing types, eMBB is characterized by large payloads and a device activation pattern that remains stable over an extended time interval. This allows the network to schedule wireless resources to the eMBB devices, such that no two eMBB devices access the same resource simultaneously. The objective of the eMBB service is to maximize the data rate. In contrast, an mMTC device is active intermittently, and uses a fixed (typically low) transmission rate in the uplink. A huge number of mMTC devices may be connected to a given Base Station (BS) but, at a given time, only an unknown (random) subset of them becomes active and attempts to send their data. Finally, URLLC transmissions are also intermittent, but the set of potential URLLC transmitters is much smaller than for mMTC. Supporting intermittent URLLC transmissions requires a combination of scheduling in order to ensure a certain amount of predictability in the available resources, thus supporting high reliability as well as random access, in order to avoid too many resources being idle due to the intermittent nature of the traffic.

Table 1 shows the measured downlink and uplink bit rates and RTT (Round-Trip Time) under various RSRP (Reference Signal Received Power) levels and three TDD schemes. TDD_1 allocates seven time slots for DL, two time slots for UL, and one flexible time slot; TDD_2 allocates six time slots for DL, three time slots for UL, and one flexible time slot; and TDD_3 allocates an equal number of time slots for DL and UL. The symbol period in TDD_1 and TDD_2 is 5 ms, while that for TDD_3 is 2.5 ms. The different RSRP values represent different sets of devices present on the factory floor, which were placed at different positions. This representation can help us to evaluate the performance under different network conditions, as well as the maximum data link speed that could be achieved via 5G. In our case, eMBB slicing is represented by the TDD allocation for large bit rate values (namely, TDD_1 7dl 2ul with low RSRP), mMTC slicing is represented by the RSRP values (as mentioned previously), and uRLLC is depicted by the lowest delay value of RTT. Based on the measurements and other parameters, a slicing schema can be recreated for an I4.0 scenario.

**Table 1.** DL/UL measurements per TDD scheme.

| | TDD_1 7dl 2ul | | | TDD_2 6dl 3ul | | | TDD_3 2dl 2ul | | |
|---|---|---|---|---|---|---|---|---|---|
| RSRP (dBm) | Downlink (Mbps) | Uplink (Mbps) | RTT (ms) | Downlink (Mbps) | Uplink (Mbps) | RTT (ms) | Downlink (Mbps) | Uplink (Mbps) | RTT (ms) |
| −75 | 137.93 | 40.61 | 12.72 | 116.42 | 63.73 | 13.45 | 91.5 | 82.79 | 12.68 |
| −90 | 124.03 | 38.39 | 13.41 | 105.09 | 48.06 | 12.53 | 82.73 | 76.76 | 12.51 |
| −100 | 113.3 | 28.59 | 13.33 | 94.61 | 35.02 | 12.9 | 74.07 | 58.37 | 12.66 |
| −110 | 54.54 | 17.66 | 12.98 | 50.15 | 22.9 | 13.23 | 37.39 | 24.34 | 11.75 |

A graphical representation of the downlink bit rate vs. RSRP for the various TDD schemes is shown in Figure 5. The uplink bit rate and the mean RTT for each TDD are shown in Figures 6 and 7, respectively. From Figure 7, we can observe that the RTT was smaller due to the shorter symbol period, which allowed for more frequent transmissions and, thus, reduced the delay.

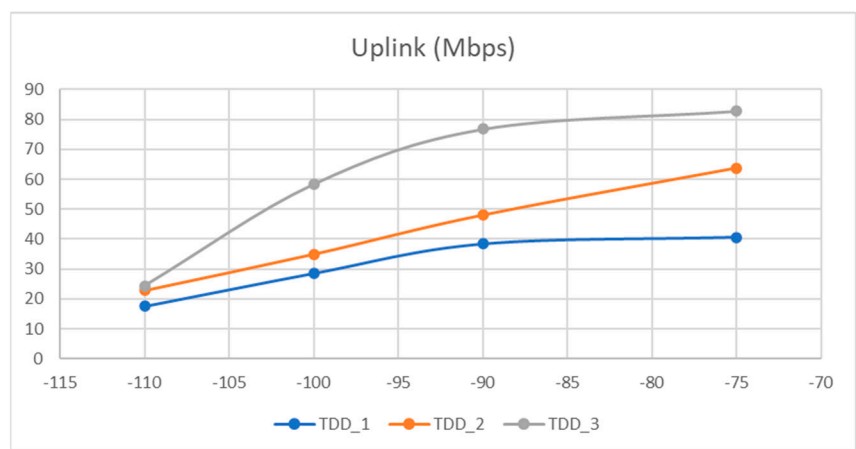

**Figure 6.** Uplink rates for different TDD values.

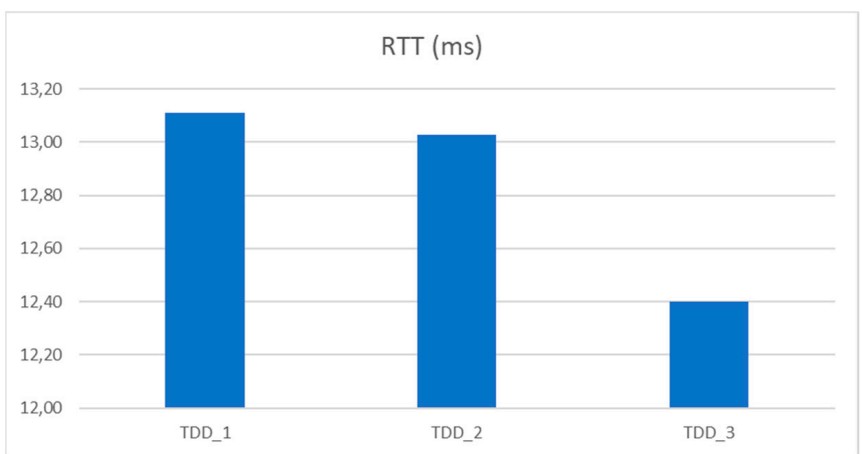

**Figure 7.** RTT measurements for different TDD schemes.

In order to compare the experimental bit rates with the theoretical ones, we used Formula (1) and its online version presented in [35]. The calculation was based on the 3GPP TS 38.306 standard: NR User Equipment (UE) radio access capabilities, and used the formula to obtain a 5G NR throughput data rate in the DL (downlink) and UL (uplink) directions. To obtain the correct result, it is necessary to enter such important parameters as the mode of the 5G network, number of aggregated carriers, number of MIMO layers, bandwidth, frequency range, modulation type, etc. For the experimental system that we used, and considering an excellent quality (−75 dBm), the theoretical bit rates were about 12–15% higher than the experimental ones for the TDD modes under consideration.

$$\text{Data rate (in Mbps)} = 10^{-6} \sum_{j=1}^{J} \left( v_{Layerz}^{(J)} \cdot Q_m^{(J)} \cdot R_{max} \cdot \frac{N_{PRB}^{BW(f),\,\mu} \cdot 12}{T_s^{\mu}} \cdot (1 - OH^{(f)}) \right) \quad (1)$$

| RSRP | Signal Strength | Description |
|---|---|---|
| $\geq -80$ dBm | Excellent | Strong signal with maximum data speeds |
| −80 dBm to −90 dBm | Good | Strong signal with good data speeds |
| −90 dBm to −100 dBm | Fair to poor | Reliable data speeds may be attained, but marginal data with dropouts is possible. When this value nears −100, performance drops drastically |
| $\leq -100$ dBm | No signal | Disconnection |

Our conclusion is that a low-cost 5G NPN may be used in an I4.0 environment to provide connectivity. The inherent flexibility of 5G makes it ideal to support a variety of machines and IoT with different requirements on the factory floor. In comparison to other methods presented in the literature, in [36], the authors considered direction-finding in mmW 5G networks by means of reported beam-based DL RSRP measurements; however, they did not report any measurements regarding the DL, UL, or RTT performance. Similar methods presented in [37,38] analyzed the error in predicting the RSRP based on the past channel measurements, and proposed a 3D UE positioning method for 5G mmW networks by exploiting DL beam-formed RS from Base Stations [39], but their results were limited to reporting of the RSRP-to-dB correlation of the 5G systems. This work extends these studies on 5G NPN networks, while providing indicators of the performance of a real-world system under factory-floor-like network conditions.

## 5. Economic and Environmental Impact of 5G NPN and I4.0

5G is expected to enable greener [39] and more manageable energy utilization [40]. 5G networks will allow for the better administration of environmentally friendly power assets, and permit families, networks, and urban communities to have more knowledge regarding their energy use and a more astute approach to oversee it. Internet of Things (IoT)-empowered items can allow for better checking and control of energy use. Joined with 5G networks, which can send data progressively, energy use can be founded on ongoing requirements in FoFs and shrewd industrial environments.

Energy efficiency is one of the major concerns when arranging and streamlining new portable networks and many related upheld methods, ranging from Base Stations to Artificial Intelligence (AI)-empowered preventive support. In this context, 5G can serve as an advertiser of measures for energy reserve funds, and as a compelling method for ensuring energy efficiency, progressively applicable in many kinds of applications [41,42]. Specifically, 5G empowers alterations to cycles and conduct, which are upheld by its high-limit, pervasive, and low-inertia organization. The thought of perspectives such as virtualization, edge processing, AI-empowered examination, and cloud storage/processing permits 5G to aid in ventures that, until recently, had to fuse techniques as basic components of energy efficiency programs, thus supporting the most productive and adaptable portion of the included assets on a case-by-case basis [43]. The canny utilization of assets can lead to lowered energy utilization in various cases—for example, support for savvy energy by executives, decreased necessity for office space and business travel, proficiency in nick-of-time supply chains empowered by prescient investigation, shrewd computerization in the development of vehicles, and conveying individualized products, among others.

Energy investment funds can be considered at three levels: the network (organization) level, the site level, and the hardware level.

- At the organization level, potential energy efficiency components include (i) flexible collaboration among 5G and LTE ranges and radios, in order to convey the perfect measure of limit with regards to a given assignment, with the optimal lowest power level; (ii) intelligent development of the board, from start to finish; (iii) progressive storage, where the information and objects that are utilized are stored near the client (perhaps in an edge figure hub), as opposed to at full scale; and (iv) the utilization of device-to-device (D2D) correspondence which, as a 5G method, takes into account availability without including Base Station equipment.
- Regarding the instances at the site level, we can recognize the accompanying components, including (i) renewable fuel hotspots for on- and off-lattice destinations, such as solar power (the expense of which has fallen by as much as 80% over the past 10 years); (ii) lithium batteries; (iii) one-site, one-bureau, and; (iv) fluid cooling, in order to lessen the energy requirements for cooling.
- At the hardware level, our potential concerns may regard, for instance, (i) efficient 5G energy enhancers; (ii) Base Station preprogrammed awake/rest periods, keeping

in mind image, channel, and transporter needs; and (iii) AI forecasting for the preemptive waking of Base Stations.

Together, the above procedures can enormously expand the energy efficiency of cell networks while lessening GHG emissions. Moreover, the device side is similarly significant, with the continual emergence of new procedures to control them effectively. The objective of 5G gadget producers and Mobile Network Operators (MNOs) is to raise battery life to 3 days for cell phones and as long as 15 years for cell IoT gadgets, in order to make certain use-cases practicable.

As Base Stations represent a high level of energy utilization, it is a basic fact that they still devour power when they are effectively taking care of information and flagging, such that MNOs must attempt to service their locales with an organization-wide energy-saving framework in mind. Classical (4G and prior) versatile networks spend only about 15–20% of their energy utilization on genuine information movement. The rest is squandered in terms of heat losses, gear continuing to run when no information is being communicated, and wasteful rectifiers, cooling frameworks, and battery units [36]. Therefore, it is imperative that new methodologies be developed to dispense with such energy wastage, or to utilize it for different purposes.

## 6. Conclusions

In this paper, we present a prototype architecture for I4.0 and 5G NPN systems, including a set of preliminary network measurement tests, focusing on network slicing and how it can facilitate factory floor integration of different OT devices. OT and IT integration is a topic with broad scope, which is currently under development, and the convergence of different technologies can accelerate the evolution towards I5.0. 5G NPNs, OPC UA, and IDS are a set of enablers that can support this process, as is clearly defined in the proposed architecture. In future steps, the I4.0 domain should focus more on trusted data sharing and novel telco technological incorporation (i.e., beyond 5G and 6G). This manuscript provides the basis for a combined 5G and I4.0, as well as experiments on the introduction of a low-cost 5G NPN for factory floors.

**Author Contributions:** Conceptualization, M.-A.K.; methodology, G.X.; software, A.O.; validation, D.S.; investigation, T.A.; data curation, A.K.; writing—original draft preparation, M.-A.K.; writing— review and editing, C.Z.; supervision, I.C.; project administration, A.K.; funding acquisition, A.K. All authors have read and agreed to the published version of the manuscript.

**Funding:** The research work presented in this article was supported by the European Commission under the Horizon 2020 Programme, through funding of the RESPOND-A project (G.A. no. 883371).

**Institutional Review Board Statement:** Not applicable.

**Informed Consent Statement:** Not applicable.

**Data Availability Statement:** Not applicable.

**Conflicts of Interest:** The authors declare no conflict of interest.

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
