# Peer review of "5G NPN Performance Evaluation for I4.0 Environments"

_applsci, doi:10.3390/app12157891_

Round 1
Reviewer 1 Report
In this research, the authors have proposed a 5G-NPN (Non-Public Network)-based Industry 4.0 convergence architecture framework focusing on manufacturing digitization and complete Information and Operational Technology (IT/OT) convergence. In the reviewer's opinion, this work is interesting and attractive but it should be improved, as follows,
1) In Section 1, the authors should mention challenging issues of 5G and Industry 4.0 such as real-time response problems, energy consumption problems, performance, privacy and security to highlight the topicality and urgency of this work. These issues were presented in the recent research in: Wireless Communication Technologies for IoT in 5G: Vision, Applications, and Challenges, Wireless Communications and Mobile Computing, 2022. The authors should reference this work.
2) The authors should add an illustration figure of "I4.0 and 5G convergence background" in Section 2 for convenience to readers.
3) In line 166, the authors write "5G ENABLED AAS", in my opinion, you re-write "5G-Enabled AAS"
4) The Equation in lines (376-377) is not numbered, the authors should re-present this equation.
5) The Equation in lines (376-377) is not numbered, the authors should re-present and focus on explaining this equation more clearly.
Author Response
Reviewer 1
1) In Section 1, the authors should mention challenging issues of 5G and Industry 4.0 such as real-time response problems, energy consumption problems, performance, privacy and security to highlight the topicality and urgency of this work. These issues were presented in the recent research in: Wireless Communication Technologies for IoT in 5G: Vision, Applications, and Challenges, Wireless Communications and Mobile Computing, 2022. The authors should reference this work.
Answer:
The authors have added a supplementary paragraph in the introductory section, addressing these topics in relation to the proposed work and added the suggested reference.
Namely: This convergence has to also address a wider set of topics related to real time re-sponse problems, energy efficiency, performance, privacy and security, topics covered in detail at [7]. In the scope of this paper, 5G-NPN deployments will be used to evaluate the real-time response criticality of I4.0 and 5G and performance. Nonetheless, security and privacy are vital components of every 5G-NPN setup and the proposed integration with IDS primitives addresses this aspect partially in the proposed work.
2) The authors should add an illustration figure of "I4.0 and 5G convergence background" in Section 2 for convenience to readers.
Answer:
The authors have created and added a relevant schema in the section, which can now clarify to the reader the set of points mentioned.
3) In line 166, the authors write "5G ENABLED AAS", in my opinion, you re-write "5G-Enabled AAS"
Answer:
The title has been corrected.
4) The Equation in lines (376-377) is not numbered, the authors should re-present this equation.
5) The Equation in lines (376-377) is not numbered, the authors should re-present and focus on explaining this equation more clearly.
Answer:
The equation has now been properly re-written and properly explained in the relevant section.
Reviewer 2 Report
This paper Presented a 5G-NPN performance evaluation for I4.0 environments.
1- The abstract should be rewritten again. The authors must include the comparison with past works and some numerical results in the abstract. Also, it should have general information about the proposed method.
2- The Introduction is too short without having enough information about the applications. I recommend to authors to add more applications in this section.
Also, the background section should be extended to new published papers for recent years with more details. There are many published works in 2 or 3 years ago about 5G telecommunications and its application. In this section should be expressed the better state-of-the-art for this application. The new references will also be examined in this part.
3- Figure 2 is not clear. I recommend modifying it to looks better and explain all details in the text.
4- There is just one formula in this article in page 9 which seems copy from other paper. Please write it correctly and explain all variable in the text.
5- Simulation conditions are not well discussed. The results are just on some specific conditions, which is not enough to draw a complete and accurate conclusion about the proposed approach. Also, the proposed algorithm should be examined on different scenarios.
6- This method should be compared with more famous methods to determine the superiority of the proposed method. The evaluations in not enough. In this paper, just one criterion has been reported for evaluations.
7- The clumsy English was highly distracting. Please, do not forget that the clarity and the good structure of a paper are important factors in the review decision. Please read the paper carefully (again) and correct it.
8- In general, the article needs much more improvement about the idea to be acceptable for publishing in Applied Sciences Journal. The authors should explain the idea in a better scientific way.
Author Response
Reviewer 2
1- The abstract should be rewritten again. The authors must include the comparison with past works and some numerical results in the abstract. Also, it should have general information about the proposed method.
Answer:
The abstract has been modified to better analyse the benefits of the paper and mentioning a example numeral value of the resulted experimental data sets.
Namely: Abstract: This paper aims to develop an open Asset Administration Shell (AAS) solution for 5G (Non-Public Network) NPN management focusing on manufacturing digitization and complete Information and Operational Technology (IT/OT) convergence. The proposed 5G NPN frame-work is evaluated in a factory simulated scenario in regard to network slicing for I4.0 and demonstrates the outlook of 5G communication in the industrial domain, achieving up to 86 Mbps of upload data rate and down to ~11ms Round-Trip-Time (RTT) for end-to-end communi-cation. The proposed framework integrates OPC UA as an enabler and middleware across differ-ent protocols, equipment and manufacturing shop floor, with the target to aggregate different industrial data and create insights on production optimization in a unified manner. The frame-work combines 5G NPNs with I4.0 environments, in the form of a complete FNMS and its respective AAS. In parallel, a set of I4.0 enablers investigated in the frame of the project, covering Time Sensitive Networks (TSN) the shop floor. The main objective of this paper is to firstly pro-pose a unified integration of various enablers in the I4.0 domain and how they all be combined with 5G technology and secondly evaluate the feasibility of hosting industrial applications and services over 5G channels via the implementation of different slicing schemas. The paper pre-sents a detailed set of experimental data sets regarding 5G downlink, uplink data links and RTT delay.
2- The Introduction is too short without having enough information about the applications. I recommend to authors to add more applications in this section.
Also, the background section should be extended to new published papers for recent years with more details. There are many published works in 2 or 3 years ago about 5G telecommunications and its application. In this section should be expressed the better state-of-the-art for this application. The new references will also be examined in this part.
Answer:
A dedicated paragraph has been added to the introduction to explore new applications in the 5G/I4.0 domain. Additionally, an overview of recent 5G communication works has been added to section 2, along with a schema better outlining the section.
3- Figure 2 is not clear. I recommend modifying it to looks better and explain all details in the text.
Answer:
Figure 2 has been modified to a higher quality one and a detailed explanation of the figure has been added to the text.
4- There is just one formula in this article in page 9 which seems copy from other paper. Please write it correctly and explain all variable in the text.
Answer:
The equation has been properly re-written and explained properly.
5- Simulation conditions are not well discussed. The results are just on some specific conditions, which is not enough to draw a complete and accurate conclusion about the proposed approach. Also, the proposed algorithm should be examined on different scenarios.
Answer:
The experimental evaluation section has been updated to better present the experimental results and better showcase the benefits of the proposed method.
6- This method should be compared with more famous methods to determine the superiority of the proposed method. The evaluations in not enough. In this paper, just one criterion has been reported for evaluations.
Answer:
In comparison to other methods proposed in the bibliography, in [36] the authors consid-ered direction-finding in mmW 5G networks by means of reported beam-based DL RSRP measurements, but the work does not report any measurements on either the DL, UL or RTT performance. Similar methods in [37], [38], analyzed the error of predicting RSRP based on the past channel measurements and proposed a 3D UE positioning method for 5G mmW networks by exploiting DL beamformed RS from Base Stations, but are limited to the reporting of the RSRP to db correlation of 5G systems. The proposed work extends this work in 5G NPN networks and actually measures the performance of a real-world system under factory floor like network conditions. This has also been added to the paper.
7- The clumsy English was highly distracting. Please, do not forget that the clarity and the good structure of a paper are important factors in the review decision. Please read the paper carefully (again) and correct it.
Answer:
The paper has been extensively reviewed and various syntax and grammar mistakes have been corrected.
8- In general, the article needs much more improvement about the idea to be acceptable for publishing in Applied Sciences Journal. The authors should explain the idea in a better scientific way.
Answer: The authors have proceeded to address and contribute further based on all the reviewers’ suggestions. The new version is better aligned to the proposed scope from the reviewers’ feedback.
Reviewer 3 Report
Manuscript numbered “applsci-1815168” has been reviewed:
The introduction needs some improvements.
Please add a section about the environmental problems of 5G.
Please add a brief paragraph about I1.0, I2.0, and I3.0 revolutions in the introduction.
It is also suggested to compare 5G, 4G, and other networks in a table and specify the superiority of 5G network's capabilities.
It is also suggested to add an economical study to the paper.
Fallowing papers are suggested for the introduction and result section:
A review of Industry 4.0 and additive manufacturing synergy
Enabling Deterministic Tasks with Multi-Access Edge Computing in 5G Networks
Integration and I4. 0 Tracking Systems for Steel Manufacturing Industry
Design and Development of a I4. 0 Engineering Education Laboratory with Virtual and Digital Technologies Based on ISO/IEC TR 23842-1 Standard Guidelines
Author Response
Reviewer 3
The introduction needs some improvements.
Answer:
The introduction has been further modified and reviewed to better showcase the main points and innovations of the proposed work.
Please add a section about the environmental problems of 5G. It is also suggested to add an economical study to the paper.
Answer:
A dedicated section has been added addressing both the environmental and the economic impact of 5G.
Please add a brief paragraph about I1.0, I2.0, and I3.0 revolutions in the introduction.
Answer:
A relevant paragraph and brief introduction between the different industrial areas has been added to the relevant section.
Following papers are suggested for the introduction and result section:
A review of Industry 4.0 and additive manufacturing synergy
Enabling Deterministic Tasks with Multi-Access Edge Computing in 5G Networks
Integration and I4. 0 Tracking Systems for Steel Manufacturing Industry
Design and Development of a I4. 0 Engineering Education Laboratory with Virtual and Digital Technologies Based on ISO/IEC TR 23842-1 Standard Guidelines
Answer:
The proposed papers have been added as references to the introductory section based on the reviewers’ feedback.

Reviewer 4 Report
This paper presents a system architecture for joint management of 5G network and production systems in I4.0 environments.
The contribution of the paper is not very clear, as an architecture is proposed but without a lot of detail. If it is to be taken as a review of existing technologies, which could be organized around the proposed architecture, many more citations and description of techniques/technologies should be done. Furthermore, the architecture is not very clearly defined; I am missing a clear delimitation of what functions and components go into each block (as said earlier, with examples of such components) and mainly the interfaces and interaction among them. Summarizing, I cannot imagine how this architecture helps to achieve the final goal of joint management.
Regarding the tests, they do not clarify any of the raised points. These measurements are just RTT measurements in different RSRP conditions. There is no clear link between this and the management architecture, nor the data sharing function proposed.
Overall, the paper is missing a clear direction and a clear description of the contributions.
Author Response
Reviewer 4
The contribution of the paper is not very clear, as an architecture is proposed but without a lot of detail. If it is to be taken as a review of existing technologies, which could be organized around the proposed architecture, many more citations and description of techniques/technologies should be done. Furthermore, the architecture is not very clearly defined; I am missing a clear delimitation of what functions and components go into each block (as said earlier, with examples of such components) and mainly the interfaces and interaction among them. Summarizing, I cannot imagine how this architecture helps to achieve the final goal of joint management.
Answer:
The paper has been thoroughly revised, new sections have been added to extend the state of the art of the related domain. The proposed architecture section has also been extended, as well as the experimental evaluation section where a detailed description of the simulation has been added. A dedicated section regarding the environmental and economic impact has also been added. The supplementary modification further clarify the objectives of the manuscript and can help the reader understand the scope and benefits of the proposed work.
Regarding the tests, they do not clarify any of the raised points. These measurements are just RTT measurements in different RSRP conditions. There is no clear link between this and the management architecture, nor the data sharing function proposed.
Answer:
The main objective of the experimental evaluation section was to simulate an industrial environment by imitating positioning and RF SNR of different devices in different locations of the factory floor. This is achieved by the RSRP parameter modification of the 5G NPN system deployed and by using different 5G terminals to measure the performance in different network conditions. The end result demonstrates the thresholds and limits of a real world 5G NPN system under different network slicing conditions and helps the reader have a clear view of the 5G landscape in the I4.0 domain.
Overall, the paper is missing a clear direction and a clear description of the contributions.
Answer:
Different sections and parts have been thoroughly revised in order to better outline the benefits and scope of the paper. The revised version includes a more extended state of the art section and more detailed explanations of the different modules and experimental evaluation tests.
Round 2
Reviewer 1 Report
This manuscript has been significantly improved compared to the previous draft. In the reviewer's opinion, it is eligible to be published.
Author Response
The authors thank the reviewer for the useful feedback and insights in the overall paper.
Reviewer 2 Report
The authors considered most of the comments and observations.
Still I insist that the authors should explain in a better way the methodology in a scientific format. This part still needs more improvement.
Author Response
The experimental section has been updated further to better explain the background of the experiments regarding 5G and the TDD scheme is was utilized.
Reviewer 4 Report
In this new revision the paper has much improved. The contribution is now more clear, and the overall direction of the paper can now be better understood. Anyway it is still a bit hard to follow at times; but a minor proofreading can fix that.
The only point that I am still lost at is on the simulations. I understand that these simulations come to proof that the use of an NPN can fulfill the requirements of I4.0 applications. And apparently these results come to show this; although maybe a more detailed discussion of the results could improve the scientific soundness of the paper. For instance, detailing how the different TDD schemes compare among each other (which is already done, albeit with very limited discussion) and how this relates to the requirements of different I4.0 applications. On that note, the paper also misses a discussion on such requirements, detailing for instance, the service types (mMTC, URLLC and eMBB) and putting numbers on their requirements for latency, reliability, coverage, etc. and examples of applications for these categories.
The other point that I think would require clarification is how these measurements, which I now understand are relevant to the paper, relate with the proposed architecture. I see the relation with the services, but not, for instance, with the data management part of the architecture.
Otherwise, the paper now is significantly improved.
Author Response
The only point that I am still lost at is on the simulations. I understand that these simulations come to proof that the use of an NPN can fulfill the requirements of I4.0 applications. And apparently these results come to show this; although maybe a more detailed discussion of the results could improve the scientific soundness of the paper. For instance, detailing how the different TDD schemes compare among each other (which is already done, albeit with very limited discussion) and how this relates to the requirements of different I4.0 applications. On that note, the paper also misses a discussion on such requirements, detailing for instance, the service types (mMTC, URLLC and eMBB) and putting numbers on their requirements for latency, reliability, coverage, etc. and examples of applications for these categories.
Answer:
The experimental section has been updated and a discrete mapping of the slicing types to the experimental parameters sets has been defined. This will help the reader clarify the correlation between the experimental outcomes and the background of 5G slicing types.
The other point that I think would require clarification is how these measurements, which I now understand are relevant to the paper, relate with the proposed architecture. I see the relation with the services, but not, for instance, with the data management part of the architecture.
Answer:
The experimental measurements focus on the performance of the 5G NPN system in a I4.0 environment. The proposed architecture involves data management and trust mechanisms, however these will be extended and implemented as-a future step and will be integrated as a backend to the 5G configuration and management of the system, as an industrial component.